# Vision encoders should be image size agnostic and task driven

## Abstract

This position paper argues that the next generation of vision encoders should be image size agnostic and task driven. The source of our inspiration is biological. Not a structural aspect of biological vision, but a behavioral trait – *efficiency*. We focus on a couple of ways in which vision in nature is efficient, but modern vision encoders not. We – humans and animals – deal with vast quantities of visual data, and need to be smart where we focus our limited energy – it depends on the task. It is our belief that vision encoders should be dynamic and the computational complexity should depend on the task at hand rather than the size of the image. We, also, provide concrete first steps towards our vision – a *proof-of-concept* solution for image classification. Despite classification being not very representative for what we are trying to achieve, it shows that our approach is feasible and promising.

## 1   Introduction

With this position paper we aim to spark a refreshed biological inspiration for computer vision models, and more specifically vision encoders. The source of our inspiration is not structural. Instead we are interested in a behavioral trait of vision in nature – *efficiency*. Why efficiency? As it is pointed out [22] a four year old child has processed more bytes of visual data than what is contained in the largest text corpora used to train modern large language models (LLMs). Reading through so much text would take a human hundreds of thousands of years. This shows that the amount of visual data we are faced with in our daily lives is enormous. We need to be very selective and smart on which parts of it to use our processing energy. A leading principle we follow in the proposal of this work is that *the goal of vision is not to process and understand every detail of what we are seeing, but to extract biologically relevant information*.

Many inventions in human history are inspired by nature, yet this often remains a point of departure. The final design being overwhelmingly shaped by engineering trade-offs and mathematical principles. This is why inspired by birds we built planes, but their wings do not flap. Convolutional neural networks (CNNs) [23] are such example. They draw inspiration from human vision, e.g. weight sharing, and hierarchical feature extraction observed in the visual cortex. Yet they perform bottom-up computation that is unlike how we process visual information in the brain. With the advent of the transformer [41], the ViT [12] architecture has become the most popular choice for visual encoders. It adds benefits like global receptive fields and the ability to quickly attend to any part of the image. However, these models are less computationally efficient – working at constant low resolution without spatially hierarchical features, with quadratic dependence on the image size due to the global self-attention between all patches.

As we think that efficiency is a crucial aspect of vision in nature, we would like to inspire future research on vision encoders that are built from the ground with efficiency in mind. In our opinion a

Submitted to 39th Conference on Neural Information Processing Systems (NeurIPS 2025). Do not distribute.

**path towards better and more efficient vision encoders, is one that focuses on models that are task-driven and image size agnostic**. Let us address these statements individually.

## 2   Why image size agnostic?

This is a common problem shared with LLMs. The amount of compute required by a model to solve a problem is much more dependent on the length of the context than on how difficult is the problem itself. We believe it is self-evident that it would be better if this dependency is flipped.

Despite significant improvement of accelerator hardware in the last years, the image sizes on which vision encoders work have not kept with the capabilities of modern cameras. A major culprit for this is that the compute in ViTs is quadratic on the number of patches / tokens. As we saw earlier the amount of visual data we are faced with is really large. Modern cameras are capable of producing images with many millions of pixels. Even linear dependence on the size of the image might not be efficient enough.

One important limitation of how we train vision encoders today, is that we process the visual data in a bottom-up fashion treating all pixels of the image equally. This is true for ViT with its fixed sized patches, but also true for CNNs applying the same filters throughout the area of the image. Image data being vast, it is also highly redundant and compressible. Compressible means that not all parts of the image bring equal value. Consider an image where the top half contains only cloudless sky. With ViT half of the patches will contain pixels from the sky. Perhaps the amount of information this part of the image brings could be contained in a single patch, or even less. This is task dependent. If you are a painter that tries to draw the scene, maybe you are interested even in minute variations of color and tone.

Our visual system is inherently selective, focusing with high resolution on only a small region of the scene. The fovea – a roughly 1–2° circle around the center of gaze – is densely packed with photoreceptors and delivers our sharpest vision. Outside this circle, visual acuity declines quickly and significantly. To construct a detailed internal representation of our surroundings, we must continually shift our gaze and refocus on successive regions of interest. We believe this contains valuable hints on how to make computer vision encoders image size agnostic. We have to extract features in a top-down manner at resolution determined by the system's *eye*, not by the size of the image. If the resolution of the image is lower than the resolution of our system, then we do not see enough detail. If the resolution of the image is higher, then it only gives us an ability to zoom. It is also important that the resolution of our system is variable – only see in high resolution small parts of the image at a time. A major contribution of this work is a complete method for extracting patches in top-down manner, independently of the image size. It is introduced in Section 5.1.

There is another consequence from how we extract patches for ViT models – a potential distribution shift of what the patches "see" when changing the size of the images after training. Let us say that for a given ViT model the patch size is 16, i.e. each patch is a $16 \times 16$ crop of the image. Now, imagine we see the same image in two different sizes $2000 \times 2000$ and $200 \times 200$. A patch in the smaller image will contain much larger portion of the image and will contain larger shapes and details. Changing the image size too much between training and testing can cause significant distribution shift for the patch tokenizer, which is usually very shallow. This is why when training ViTs it is very important to use random resized crop augmentation [40]. This is problem that exists for CNNs as well. This distribution shift happens because we extract features in a bottom-up fashion, starting from the pixels. If the image gets resized, the types of detail each layer of ViT or CNN networks sees changes accordingly. See a visual example of this distribution shift in Figure 4.

## 3   Why task driven?

Now, to our second point – why vision encoders should be task driven as opposed to task-agnostic. *Faced with vast amounts of visual data what we do to extract the biologically relevant information is basically compression. Different tasks require compressing the data in different ways.* We cannot

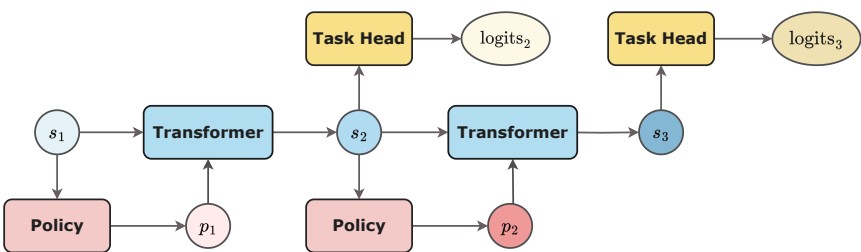

Figure 1: Overall system architecture. We evolve an initial state (prompt) through iteratively calling the same transformer model with different patches. The patches to extract are selected by the policy based on the current state. The task head produces task-specific output given an internal state. We have a task output after each iteration.

expect to classify an image and find Waldo [1] with the same computation. The two tasks require vastly different approaches and computational resources.

In our opinion, using task agnostic encoders is not only a case of inefficiency, but a fundamental limitation on how we use vision models today. Imagine playing a game where you are given an image for a second, then the image is hidden and you are asked questions about its content. What was the image of? Were there people? How many? What is the color of the clothes of the leftmost person? What about their shoes? This game is very difficult. Asked about a small detail after the picture is hidden would make one struggle a lot. It might be impossible if attention was not paid to this particular detail. But if the question came before looking at the image, it becomes trivial. Handling our daily lives this way seems unimaginable, yet this is what we expect from computer vision encoders. Typically, when we use a vision encoder in a system, say a vision language model (VLM), we run the input image through the encoder, extract features from it and use that features downstream. We always extract the same features, no matter what the downstream tasks is. This is a lot like playing the game above. What if we process the text prompt first and use it to guide us how to look at the image.

Since we already want our encoders to be image size agnostic, it provides an opportunity to make the encoder task driven as well. If an encoder is going to be image size agnostic it will have to work with limited data. If it works with limited data, then this naturally calls for an iterative process. An iterative process can be guided in a task-aware way.

# 4 Proof-of-concept solution

With this paper we do not limit ourselves to just advocating that vision encoders should be task driven and image size agnostic. We are going to propose a concrete system (combination of models) that can act as such an encoder. We also provide a concrete implementation which we test on the image classification task on Imagenet-1K [10]. While this task alone is not enough to showcase task driven abilities, it is an established benchmark to show that our ideas are feasible.

Similarly to other biologically inspired works in computer vision, we are also trying to emulate the benefits of fovean vision in our eyes and brains. We are not trying to structurally copy this model. Our goal is to propose a computational paradigm in which the efficiency benefits of fovean vision are easily achievable. We are using a transformer [41] as our main building block.

Since we have to work with small contexts, the main idea is to use a transformer iteratively. At each step the transformer processes a small set of image patches, while evolving an internal state that could also be referred to as memory. The input patches are extracted from the image in a top-down manner, allowing us to be image size agnostic. The final component of our proposal is a policy which given the internal state of the transformer decides which patches to extract next, i.e. where to look. Note, that this computational system is not new. Previous works like [34, 29, 1] have attempted to

---

[1]"Where's Waldo?" is a children's puzzle book by Martin Handford, Little, Brown and Company, 1987. "Where's Waldo?" is a registered trademark of Candlewick Press.

Multi-zoom patches

Patches

Figure 2: Extracting multi-zoom patches – patches with the same center but varying sizes. Notice that the area of the crop outside the image is padded with 0s.

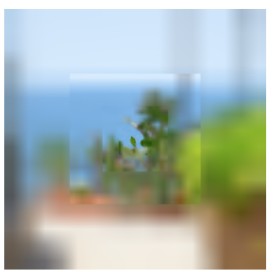

Figure 3: Example multi-zoom patches overlayed over each other after being resized to their original size. The center here is sharp from the patches with significant zoom. The top level patch covers the whole image, but it is of very low acuity.

build computer vision models in a similar way. In fact a major goal of this paper is to revive interest in these works with modern architectural components.

Figure 1 shows the schematic view of the system. The main building blocks are a transformer, a policy, and a task head. In the context of this paper the task head is a simple MLP module that produces logits for the classification task. We start with a learned initial state $s_1$. In a task-driven system this initial state could be treated as a task prompt. The policy module takes the state $s_1$ and selects the patches $p_1$ to extract. The transformer is given the state and the patches, and returns an updated state $s_2$. Now we can run the task head on $s_2$ and potentially end the computation. Alternatively, we can use the policy on $s_2$ to get new patches $p_2$. The transformer takes the new patches and updates the state to $s_3$. This process repeats and can continue as long as the task requires.

## 5 Implementation for image classification

In this section we provide concrete implementation for all the building blocks of our proposed system. All components are built within the context of image classification, but are applicable beyond this task. We show three things

- how to extract patches in top-down manner,
- how to train an iterative transformer with evolving internal state,
- how to train a policy to tell us where to look next.

The first two are valuable contributions of this paper, the last one requires further work.

### 5.1 Top-down patch extraction

The transformer is a general computational block. It does not have to work only with patches of fixed size. The only requirement is that all tokens are of the same dimension. We allow any square crop of an image to be a patch. It only needs to be resized to a predefined fixed size that a tokenizer (or `PatchEmbed` module) accepts. In our experiments this size is $16 \times 16$. This matches well our fovean vision. If the crop is a small square in the image, then the change to the resized $16 \times 16$ version will be minimal, i.e. the patch is of high resolution, but cover small portion of the image. If the crop is a large square, then the resized version will be significantly different (a lot of detail will be lost), i.e. the patch is of low resolution, but covers large portion of the image.

**Multi-zoom patches**. The definition above gives us a very flexible notion of what a patch is. However, in order to be computationally efficient we want to extract patches from images in a systematic way

which is scalable on modern highly parallel computer architectures. As mentioned a patch can be any square crop. Let us say the size of the crop is $C \times C$. We define a patch using three coordinates $(x, y, z)$. $(x, y) \in [0 - 1]^2$ are the coordinates of the center of the crop. This is in relative coordinates, i.e. both values are between 0 and 1. $z$ is a non-negative number that represents the zoom level of the crop. If $z = 0$, then the size of the crop $C = min(H, W)$. When $z > 0$, then the size of the crop is $C = min(H, W)/2^z$. Here, $H$ and $W$ are the height and width of the image respectively. This is the essence of our top-down approach. The size of the patch is relative of the image size. If the image size changes a patch remains a crop of the same part of the image, but its size is pixels will differ.

To make extracting patches more efficient and parallelizable on modern hardware, we always extract a fixed sequence of patches from a given center $(x, y)$. In other words given a patch center $(x, y)$ we extract $M$ patches, one for each value of $z = \text{jnp.linspace}(0, \max_z, M)$. This way, for a given gaze location $(x, y)$ we get a series of patches of decreasing resolution and increasing image coverage. See extracted patches in this manner in Figure 2. This is similar to how our eyes work. We see only a small circle around the center of gaze sharply. Our field of view is large but of low acuity. Figure 3 shows multi-zoom patches overlayed over each other. The patches are first resized to their original size from $16 \times 16$. This is why the largest patch is very blurry. Note that this is not what the transformer see. This is only drawn for us to appreciate the idea. The patches that the transformer sees are the ones in Figure 2. In Appendix A.1 we provide implementation for efficient multi-zoom patch extraction in Jax. Our multi-zoom patches are similar to the ones used in [29]. They also extract series of patches from the same center each twice the dimensions of the previous. However, a crucial difference is that there the approach is bottom-up. We extract the patches in top-down manner and hence we are image size agnostic.

## 5.2 Iterative transformer with internal state

Multi-zoom patches gives us a size agnostic way to extract patches from an image, but there is one problem – there are infinitely many such patches, especially when $x$ and $y$ are floating numbers. With ViT we have a fixed set of patches that go through the transformer. The idea of the multi-zoom patches is to use them iteratively, similarly to how our vision works. Instead of taking all the image contents at once, we iteratively gaze at different parts of the image and build our understanding of what we are seeing.

How can a transformer evolve an internal state? The most popular vision transformer ViT [12] is based on BERT [11]. There a sequence of input tokens (patches) and a learned CLS token get transformed with full self-attention between them. The CLS token can be treated as an internal state. Inspired by the success of the DETR [4] transformer decoder with learned $N$ object queries, we try to expand the internal state from a single vector (the CLS token) to a set of $N$ embeddings.

The transformer we use is basically the same as ViT with registers [9]. However, instead of keeping the transformed patches and discarding the registers, we do the opposite. The input to the transformer is a set of patches and $N$ learned state vectors. After we run the transformer, we discard the transformed patches, and keep the evolved state, which we then use as input for the next iteration with different patches.

Figure 5 shows the application of the transformer. It is important to note that while the output state is input for the next iteration, we do not train our model as a recurrent neural network (RNN). We do not allow gradients to backpropagate between different iterations. Our solution is similar to the Recurrent Memory Transformer [2]. The main difference is that we do not perform backpropagation through time [33, 30, 43].

## 5.3 Learned policy for where to look

The last part remaining is figuring out where to look. In this prototyping stage, we use reinforcement learning through gradient based policy optimization [44]. More specifically we use the Group Relative Policy Optimization [36] (GRPO) algorithm. With multi-zoom patches we extract series of patches with the same center $(x, y)$. This means that we can model our policy with continuous actions. Concretely, the internal state of the main transformer is used as observation input to the policy, which selects the next gaze location $(x, y)$.

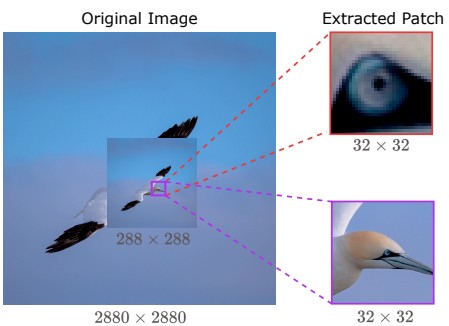

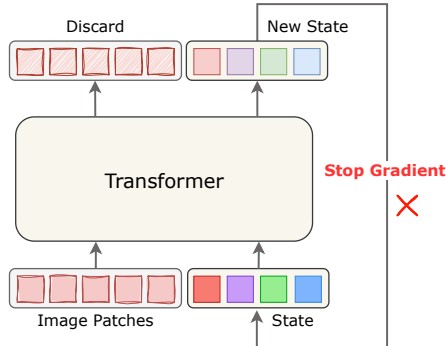

Figure 4: Example of distribution shift with ViT patches when changing the image size. The original size of this image is $2880 \times 2880$. On the right we see two patches (of size 32) extracted from a resized image ($288 \times 288$) and from the original image. You can see how much types of details visible change due to resizing.

Figure 5: Schematic view of how we keep an evolving internal state with the transformer. It processes patch tokens together with the previous state as input. After multiple self-attention layers between the input we get updated state together with updates patches than are discarded. In the following iteration they will be replaced by different ones.

With this approach one of main challenges is how to train the transformer and the policy end-to-end. The internal state of the transformer is the visible observation for the policy. If it changes during training it is very difficult to keep the policy relevant. Additionally, the training loops and dynamics for training the transformer and the policy using policy optimization are quite different. At this proof-of-concept stage we train the system in two stages. First, we pretrain the transformer and the task head using random policy, i.e. at each iteration choose a gaze center uniformly at random. In the second stage freeze the transformer and optimize the policy using GRPO. More details on our approach can be found in Appendix A.2.

## 6 Classification results

Image classification might not be the most suitable benchmark for task-driven vision encoders. However it provides a good reference for testing our solution as a proof-of-concept. With it we are able to validate the feasibility of our build blocks. The main aspects we want to test are the following:

- Multi-zoom patches are suitable input for vision transformers.
- It is possible to train iterative transformers that see only part of the content in each run and evolve an internal state.
- It is possible to teach a policy where to look in order to solve the image classification problem.

All experiments we performed are on the ImageNet-1K [10] datasets. We skipped training on datasets with smaller images like MNIST [23] and CIFAR [20] because with very small images it is hard to test the image size agnostic part of the encoder. All patches will contain significant portion of the image. Implementation details and training setup for the experiments can be found in Appendix A.3.

### 6.1 Iterative transformer with ViT patches

First, let us test the ability of iteratively running a transformer with an evolving state. As a baseline we train a ViT-Base model with patch size 16. The size of the input images is $256 \times 256$, so with each patch being $16 \times 16$ we end with a total number of patches 256. After that we shuffle the patches and split them into 4 random groups, each containing 64 patches. Then we train our transformer with episode length of 4 – in each episode it sees one of the groups of patches. The internal state contains $N = 32$ vectors. You can see the results of this experiment in Table 1. Note, that while we are training with multiple iterations there is no gradient back propagation between steps. At each step

Table 1: Results from iteratively training a transformer with shuffled groups of ViT patches.

| Metric | Shuffled Step 1 | Shuffled Step 2 | Shuffled Step 3 | Shuffled Step 4 | ViT |
|---|---|---|---|---|---|
| Acc @ Top 1 | 0.66 | 0.69 | 0.71 | 0.72 | 0.78 |
| Acc @ Top 5 | 0.86 | 0.88 | 0.89 | 0.90 | 0.93 |

the input state is detached from the previous. These results are very promising. We do not expect this approach to match the performance of ViT in terms of accuracy. We work with the same overall data (the same patches), but do not allow full self-attention between all patches. Only randomly selected patches are allowed to self attend to each other. Then the result of this self-attention needs to be stored in shared free-form internal state. These results show that this state works well as a form of memory. Note that in all epochs during training the validation accuracy at step $i$ is always higher than the validation accuracy at step $j$, when $i > j$.

## 6.2 GRPO policy with multi-zoom patches

The results from Section 6.1 are promising, but what we are really interested in is combining the iterative transformer with the multi-zoom patches and a learned policy to tell us where to look. We use the following setup. At each step the transformer sees $M = 16$ multi-zoom patches centered at a particular point $(x, y)$. The internal state of the transformer is $N = 16$ vectors, i.e. the total number of input tokens to the transformer is 32. We train with episode length of 8. First, we pretrain the transformer and the task head using a random policy, i.e. for each episode we choose the patch center coordinates uniformly at random between 0 and 1. Then we freeze the transformer and task head, and train a policy with GRPO. See the performance in Table 2. The policy does helps us to perform well. However, the results with the random policy show why image classification might not be the best task for this test. Even a random policy has reasonable performance. Note that with multi-zoom patches we see large portion of the image in low resolution at each step. Often this is enough to get a good sense of what might be in the image. While these results are quite satisfactory as a proof-of-concept, we believe future research in this area could improve the performance significantly. A lot of questions related to how to train a good policy in a general sense remain open as discussed in Section 7.

Table 2: Performance of GRPO policy with multi-zoom patches.

| Model Variant | Acc @ Top 1 | Acc @ Top 5 |
|---|---|---|
| Pretrain Rand Policy - Step 1 | 0.36 | 0.58 |
| Pretrain Rand Policy - Step 4 | 0.55 | 0.78 |
| Pretrain Rand Policy - Step 8 | 0.60 | 0.82 |
| With Policy - Step 1 | 0.51 | 0.74 |
| With Policy - Step 4 | 0.62 | 0.83 |
| With Policy - Step 8 | 0.65 | 0.85 |

## 7 Open questions

Our experiments show that the multi-zoom patches are easy to handle by the shallow patch tokenizers used in modern ViTs. This is despite the fact that they represent image data in varying scale, from tiny crops to almost the whole image. Additionally, we saw that training a transformer iteratively with an evolving internal state is quite manageable. The part that contains open questions is the policy, i.e. learning where to look.

**End-to-end training of the whole system.** With this work we use a two stage approach for training our system on image classification. First, we trained the transformer using random selection of multi-zoom patches. Then with a fixed transformer we trained the policy with GRPO [36]. This approach works because for image classification training the transformer with randomly selected patches is reasonable. For other more complex tasks this might not be the case. As far as we are aware it is an open question how to train the transformer and the policy that uses the transformer's state as observation in a single training loop. While the transformer is training, the distribution of the

state vectors will be changing. With this the input observations for the policy will be changing as well. On top of that, the training dynamics and the training loop for reinforcement learning algorithms like PPO [35] and GRPO [36] are very different from the ones for training a transformer with supervised or self-supervised learning.

**Large scale self-supervised pretraining.** We strongly believe that powerful and general vision encoders should be pretrained in a self-supervised manner or large amounts of data. The question about end-to-end training of the transformer and policy is still valid. However, there are additional open questions. For example, what is the goal of the policy while pretraining. Let us say we are training a vision model in self-supervised way, e.g. training with the DINO [5] objective, but given multiple steps to look at different locations. What is the goal of the policy in this case? How should we compute the rewards for individual actions. Additionally, we want to train a task-driven encoder, but we are pretraining on a single self-supervised task which will not be encountered after the pretraining stage.

**Should the policy be trained with reinforcement learning?** This is another interesting question. If there is a way to make training the policy differentiable we might be able to train the system end-to-end. A promising approach here might be based on implicit neural representation of images [28, 46]. We are not aware of successful attempts to use reinforcement learning during large scale pretraining.

**Large scale pretraining only for the transformer.** Another option might be to pretrain only the transformer with a fixed or random policy. Then add the policy aspect only when finetuning for particular tasks. Such solution might be in conflict with the task driven property we desire.

# 8 Related work

## 8.1 Transformers with hierarchical spatial features

Before the advent of the transformer based architectures, vision encoders were primarily based on convolutional neural networks [23, 21, 37, 38, 16]. The models work with spatial features of varying resolution. Perhaps the most popular transformer based architecture at the moment is ViT [12]. It is a pure transformer backbone that splits the image into equal sized patches (e.g. $16 \times 16$) and treats them as tokens. It does not work with hierarchical spatial information, but at a constant low resolution. It also scales quadratically with the number of pixels in the image due to the self attention mechanism between all tokens. DeiT [39] introduces data efficient training, but keeps the architecture unchanged. PVT [42] uses a transformer backbone for dense prediction tasks with hierarchical features inheriting advantages from both CNNs and ViT. Swin [25, 24] takes a different approach by keeping the same sized patches from ViT by limiting the self-attention with local windows. MViT [13] proposes transformers with multi-scale feature pyramid. PiT [17] introduces pooling-based vision transformer. Hybrids between convolutional and transformer based architectures exist as well, like LeViT [14]. Cross-ViT [6] introduces a transformer that handles different resolutions with separate parallel branches. Twin-transformers [7] utilize spatially separable attention mechanisms which consists of two types of attention – local self-attention and global subsampled attention. Focal transformers [47] introduce self-attention for local-global interactions. All these methods improve the efficiency of the ViT architecture, however in our opinion these improvements can go further. Even linear dependence on the image size is expensive, since images generated from modern cameras contain many millions of pixels. Additionally all of them still extract features in a bottom-up fashion, being prone to distribution shift when changing the image size drastically. Changing the size of the same image will lead to all layers of the networks to deal with different level of detail, even CNN architectures.

## 8.2 Methods inspired by fovean vision

The fovea is a small part of the retina of our eye where the concentration of light receptors is very high. A $1 - 2°$ circle around the direction of the gaze where the resolution is highest. As we go away from the center of gaze the resolution drops quickly. We have to constantly move our eyes, a process called saccades, to gather details from various parts of the scene. This process is highly efficient and flexible. It has been a source of inspiration within the computer vision community for a long time. [34] is a seminal work on fovean inspired vision models. It also uses a learned policy, however through a learned world model. [29, 1] propose a method for vision models that is very similar to

ours. They also use reinforcement learning to train a policy to direct the next glimpse. They also use multi-resolution patches that are similar to ours. A few key difference to our approach is that their multi-resolution patches are bottom-up instead of top-down, and that their model is trained as a recurrent neural network (specifically LSTM [18]). One of the main goals of our position paper is to inspire us to revisit these older works with modern architectural components. [27] proposes a more direct approach to transform or sample the input image in a way that mimics the retina. Then the transformer image is sent to a CNN. [19] combines a CNN backbone with foveation pooling mechanism and a transformer. [3] proposes a combination between bottom-up saliency detection and top-down attention using unsupervised learning for object detection.

### 8.3 Iterative transformers with evolving internal state

A key component of the method we want to propose is using a transformer iteratively with limited context while it evolves an internal state. In essence treating the transformer like RNN, but without propagated gradients between iterations. The internal state can be considered memory of what the transformer has seen so far. Transformer-XL [8] introduces segment-level recurrence for transformers. Effectively caching the hidden states from the previous segment, allowing to double the length of the context without backpropagating gradients into it. The Compressive Transformer [32] builds on top Transformer-XL. Instead of discarding old cached hidden states it compresses them. GMAT [15] adds dedicated learned memory tokens. It still uses long context by applying sparse self-attention within the context, and dense attention is performed by the memory tokens only. Memformer [45] adds an external dynamic memory to encode and retrieve past information. They also do backpropagation through time over very long sequences. The Recurrent Memory Transformer [2] add memory as learnable tokens appended to the input context. Their solution is similar to the one we propose. A crucial difference is that they do backpropagation through time [33, 30, 43].

## 9    Alternative view

It is our view that having vision encoders to be image size agnostic is self-evidently desirable. Note that this does not mean that the model is completely independent of the image size. Strictly following this might be impossible. The main idea behind this statement is that the computational requirements of a vision encoder should not be increased by increasing the size of the image, unless the task itself gets more difficult with the increase of resolution. Such examples may include camouflaged object detection, or trying to retrieve all the text in an image. Increasing the resolution might make a lot more text available to parse. As such we do not provide alternative view for the desired property of being image size agnostic.

Things are a bit more nuanced with the task driven property. In our opinion the game where the image is hidden before asking question about it, is a good example why task driven encoders are desirable. However, there is merit in encoders being task agnostic. Modern models like DINOv2 [31] are task-agnostic and perform very well on multitude of tasks. Their strong advantage is that they are very easy to use. One can easily use DINO as a backbone in any system that requires visual perception. In our view task driven encoders are going to be eventually better and more efficient than task-agnostic ones. However, reaching this point is going to be challenging. Similarly to self-supervised and supervised models. For a long time it was believed that self-supervised models are better, but their performance did not match that of supervised ones. Today models pretrained in self-supervised manner outperform supervised ones on most of the benchmarks.

## 10    Conclusion

In this paper we argued that the future of vision encoders should be focused on models that are image size agnostic and task driven. We made multiple biological references to support our position. We also showed a proof-of-concept system that can be used as such an encoder. We also showed that this line of research is not new  [34, 29, 1]. It is our hope to inspire us to revisit these ideas with modern architectural components. We also provided valuable research contributions to this cause – top-down manner of extracting multi-zoom patch in image size agnostic way, and iterative transformer capable of evolving internal state without backpropagation through time.

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

# A  Technical Appendices and Supplementary Material

The supplementary material is organized in the followin way. Appendix A.1 provides efficient code in *Jax* to extract multi-zoom patches. Appendix A.2 contains details on how we use GRPO to train a policy where to look. Finally, Appendix A.3 contains implementation details for our *proof-of-concept* solution for image classification.

## A.1  Extracting multi-zoom patches

For completeness we provide efficient code in Jax for extracting the multi-zoom patches in a top-down manner. Since we extract patches of consistent sizes the extraction process can be easily parallelized. See Algorithm 1. The function itself can be used with `jax.vmap` to parallelize the computation for a whole batch of images.

---

**Algorithm 1** Function to extract multi-zoom patches for an image.

```
def extract_patches(image, center, patch_size, num_patches, max_z):
    height, width = image.shape[:2]
    # Get the zoom levels
    zs = jnp.linspace(0, max_z, num_patches)

    # Center of each patch in pixels
    center_y = center[1] * height
    center_x = center[0] * width

    def extract_single_patch(z):
        """Extract for a given zoom level."""
        # Patch size in the image space
        patch_img_size = min(width, height) / 2**z
        scale_factor = patch_size / patch_img_size
        translate_x = patch_size / 2 - scale_factor * center_x
        translate_y = patch_size / 2 - scale_factor * center_y
        return jax.image.scale_and_translate(
            image,
            shape=(patch_size, patch_size, 3),
            spatial_dims=(0, 1),
            scale=jnp.array([scale_factor, scale_factor]),
            translation=jnp.array([translate_y, translate_x]),
            method="bilinear",
        )

    # Batch extract for all zoom levels
    return jax.vmap(extract_single_patch)(zs)
```

---

## A.2  Details on learning where to look with GRPO

Once we have trained the transformer and the task head with random gaze locations it is time to teach a policy where to look. We use the group relative policy optimization algorithm (GRPO) [36]. The internal state from the transformer is the state observation visible to the policy. The rewards comes from how quickly we decrease the cross entropy loss (since we only work with classification here). Given an input image $i$, we use an old version of the policy $\pi_{\theta_{old}}$ to collect $G$ traces $\{o_1, ..., o_G\}$. A trace $o_i$ consists of $n$ actions $o_i = (s_1, a_1, s_2, ..., a_n, s_{n+1})$. We optimize the following objective:

$$\mathcal{L}(\theta) = \mathbb{E}[i \sim P(I), o \sim \pi_{\theta_{old}}(O|i)]$$

$$\frac{1}{G}\sum_{i=1}^{G}\frac{1}{|o_i|}\sum_{t=1}^{|o_i|}\left\{\min\left[\frac{\pi_\theta(a_{i,t}|s_{i,t})}{\pi_{\theta_{old}}(a_{i,t}|s_{i,t})}\hat{A}_{i,t}, \text{clip}\left(\frac{\pi_\theta(a_{i,t}|s_{i,t})}{\pi_{\theta_{old}}(a_{i,t}|s_{i,t})}1-\epsilon, 1+\epsilon\right)\hat{A}_{i,t}\right]\right\}. \quad (1)$$

$\hat{A}_{i,t}$ are the group normalized advantages. To understand how they are computed, we first need to define how to get the rewards. We employ two types of rewards. The first one is an end of the episode reward shared across all actions. It is based on the loss after the end of the episode. The second one is an immediate reward for each action defined by the reduction of the task loss. With the first approach the unnormalized advantages $\alpha_{i,t}$ are defined as,

$$\alpha_{i,t} = \log \mathbb{P}\left(\tau(s_{i,n+1}) = y\right). \quad (2)$$

$\tau(s)$ is the output of the task head and $y$ is the ground truth label. In essence, the reward for each action is the negative cross entropy loss, computed at the end of the trace, i.e. with the task head output from the last state $s_{n+1}$.

With the second approach the un-normalized advantage $\alpha_{i,t}$ is defined as,

$$\alpha_{i,t} = \frac{l_{i,t} - l_{i,t+1}}{l_{i,t} + l_{i,t+1}}, \tag{3}$$

where $l_{i,t}$ is the classification loss after applying the task head to state $s_{i,t}$, i.e. $l_{i,t} = -\log \mathbb{P}(\tau(s_{i,t}) = y)$. This is basically the improvement ratio of the loss based on the current action. We have both the current and the next loss in the denominator to keep the reward symmetric around 0. The final group normalize advantages are defined as,

$$\hat{A}_{i,t} = \frac{\alpha_{i,t} - \bar{\alpha}_t}{\hat{\sigma}(\alpha_t)}. \tag{4}$$

Here $\bar{\alpha}_t$ is the mean and $\hat{\sigma}(\alpha_t)$ is the standard deviation of $\alpha_{.,t}$. Note, that we normalize across the traces in the group, but separately for each time step. This is particularly important for the second approach for the advantages using only the immediate reward. The reward during the early steps is typically much higher then that for the latter steps. This is because we start with very low confidence about what is the class of the image, but once it is high it is much harder to increase further. This is why we need to normalize separately for each time step.

Both ways of defining the reward have their pros and cons. With an end of episode reward shared across all actions we attribute the same reward for each action. However, some actions are good and some actions are not good in the same trace. There are specific challenges with image classification. It is a task where a random policy performs very well, and bad actions along the way (looking at uninformative place) does not hurt the performance. This is also the reason we use GRPO instead of Proximal Policy Optimization (PPO) [35]. It is challenging to train a good critic when from any state a couple of good actions will lead to a low loss. The second approach for computing the loss is more direct – rewarding actions based on their immediate contribution to decreasing the loss. The challenge here is normalizing the advantages. The starting point for each action is different and thus it is hard to fairly normalize the advantages.

## A.3 Training setup and implemenation details

All our experiments are performed on the ImageNet-1k [10] dataset on TPU v4-32 machines.

### A.3.1 Transformer

For the main transformer we use implementation similar to the ViT implementation of DINOv2 [31]. The model we use is compatible in size with the ViT-Base models. It consists of 12 layers. The embedding dimension of each token is 768 (spread across 12 heads). The input to the transformer contains $M$ patch tokens and $N$ state query vectors. If it is the first iteration, the $N$ state query vectors are the learned task prompt. In subsequent runs the $N$ state query vectors are the output state from the previous run. Similarly to DETR [4] the state query vectors are added to each layer of the transformer as skip connections [16]. The $M$ token inputs are the tokenized multi-zoom patches centered at a single location, summed with their respective positional embeddings. The position embeddings are defined on a three dimensional input $(x, y, z)$, where all values lie in the interval $[0 - 1]$. The positional embeddings are computed with a small MLP network with 3 inputs and 768 outputs. Note that the zoom level $z$ is also a value between 0 and 1. The actual values when extracting the patches are between 0 and $Z_{max}$, but they are scaled to be in the interval $[0 - 1]$ so that the input to the positional embedding module is normalized. All of the $M$ tokens are centered around a fixed center $(x, y)$.

### A.3.2 Task head

Since the only task we use in our experiments is classification we have a simple head that combines the $N$ states vectors through a learned linear combination. Then a simple MLP module is used to output $K$ logits, where $K$ is the number of classes.

### A.3.3 Policy

Since we extract MZ patches with the same fixed center, the policy's action can be described only with the $(x, y)$ coordinates. Hence, we use continuous actions, returning a tuple of values between $0$ and $1$. We use a DETR-like transformer to parametrize the action distribution which is a mixture of $K$ Gaussians. The transformer contains $K + 1$ learned query vectors (one for each mixture and one for the categorical distribution to select which Gaussian to sample from) and does cross-attention to the $N$ state vectors. Then with a simple head we extract the mean coordinates for each Gaussian of the mixture. The standard deviation is a fixed parameter during training. We use deterministic actions during inference. We opted against representing the action with a single Gaussian, because this assumes there is a single good action from each state. This is clearly not true with the classification task.

### A.3.4 Training stages

As mentioned we train the whole system in two distinct stages. In stage 1 we only train the transformer with the task head using a random policy. Each episode contains 8 steps and in each step we select the center $(x, y)$ uniformly at random. Training is done for 300 epochs with batch size of 1024. The learning rate is $5 \times 10^{-4}$ decayed with cosine schedule [26], after a linear warmup. In this stage we also use MixUp [48] augmentation.

In stage 2, the trained transformer and task head are frozen. And we only train the policy. We do $4$ epochs over the whole ImageNet dataset with batch size $4096$. For each batch we collect $G = 16$ traces of length $8$. This is the data for the inner epochs for GRPO. We optimize the GRPO objective for $8$ inner epochs for each batch.

