# OpenReview forum: "Vision encoders should be image size agnostic and task driven"
_NeurIPS.cc/2025/Position_Paper_Track — Submitted to NeurIPS 2025 Position Paper Track_

### Official Review · Reviewer_peVm · 2025-08-05

**Significance:** 3
**Presentation:** 2
**Rating:** 3
**Confidence:** 4

**Summary:**

This manuscript claims the form the vision encoder should be. The authors propose two properties, image-size agnostic and task-driven, which are motivated by their opinion with biological inspiration. Pilot experiments are performed as a proof-of-concept.

**Strengths:**

- The concept based on the biological motivation is worthy.

- I agree that the larger image size, such as 2000x2000, would become a critical issue when using ViT with standard patch partitioning.

**Weaknesses:**

- Evaluation of the idea - I agree with the "image size agnostic" part; patch partitioning fashion in the ViT is unnatural for larger image sizes. However, I think opinions in the research community will vary on the "task-driven" part. The current approach, where the foundation model is task agnostic and only the final head is task specific, would be thought of as sufficient.

- Although I understand that this manuscript is for the position paper track, I think that it needs more objective evidence for the two claims than just opinion based on verbal statements. In particular, the task-driven part lacks solid references or relevant data on biological motivation.

- Although the term encoder is used, it seems like the authors are tackling the tokenization step of an image, such as patch embedding, rather than the encoder itself. In other words, the topic is rather related to image representation, and the encoder itself is still ViT.

- The top-1 accuracy in Table 2 is too low. I think a reasonable baseline would be 0.7 to 0.8.

**Questions:**

See the weaknesses above. Overall, I understand that this is for the position paper track, but the two claims require more solid evidence rather than opinions.

**Alternative Position:**

Yes, and alternative positions are well-considered and addressed by the argument

**Author Identification:**

No.

**Context:**

3

**Discussion:**

3

**Ethics:**

["NO or VERY MINOR ethics concerns only"]

**Position:**

Yes, the paper argues for or against a position related to machine learning.

**Support:**

1

**Thoroughness:**

4

---

### Official Review · Reviewer_mgPM · 2025-08-10

**Significance:** 4
**Presentation:** 4
**Rating:** 7
**Confidence:** 4

**Summary:**

This paper advocates for vision encoders to be "image size agnostic" and "task-driven," drawing inspiration from the biological vision. The authors support this position by by introducing an iterative transformer system that processes images by selectively extracting "multi-zoom patches" in a top-down manner, guided by a learned policy. This system maintains an evolving internal state as a form of memory, allowing it to build an understanding of the image over several steps. They demonstrate the feasibility of their approach on the ImageNet-1K image classification task, with a two-stage training process. The position, ultimately aims to reignite interest in biologically-inspired vision models with the idea of making them more efficient and task-aware.

**Strengths:**

- The paper is well-written, logically structured, and easy to understand. It uses effective analogies and illustrations
- The overall idea is quite interesting. Also, it pinpoints a fundamental inefficiency in current vision encoders regarding their image size dependency.
- This work offers a specific, implementable proof-of-concept, rather than just a hypothesis. The results on ImageNet-1K are promising, and demonstrates feasibility of the idea

**Weaknesses:**

While the position effectively argues for image size agnosticism and task-driven encoders, the primary focus on these two aspects for designing/proposing a biologically-inspired vision encoder might be in a way, limiting, especially if we consider real-world visual complexities. Real-world environments present a multitude of other challenges beyond just image resolution, such as occlusions, lighting variations, background clutter, object poses, etc. Although the proposed method could implicitly benefit these scenarios by focusing on salient features, the paper doesn't explicitly discuss them. Expanding on this would further solidify the breadth of the proposed approach's potential impact.

**Questions:**

See weaknesses

**Alternative Position:**

Yes, and alternative positions are well-considered and addressed by the argument

**Author Identification:**

No.

**Context:**

4

**Discussion:**

4

**Ethics:**

["NO or VERY MINOR ethics concerns only"]

**Position:**

Yes, the paper argues for or against a position related to machine learning.

**Support:**

3

**Thoroughness:**

4

---

### Official Review · Reviewer_X9ha · 2025-08-11

**Significance:** 3
**Presentation:** 3
**Rating:** 6
**Confidence:** 3

**Summary:**

The authors propose that current vision architectures (e.g., ViTs, CNNs) process images inefficiently because computational complexity scales with image size rather than task difficulty. They advocate for a shift toward models whose computation depends on the task at hand, processing high-resolution information only where needed—similar to human foveal vision. They outline a proof-of-concept: a **top-down, multi-zoom patch extraction method**, an **iterative transformer with evolving internal state**, and a **policy module** (trained via GRPO) to decide where to "look" next. Experiments on ImageNet-1K validate feasibility, though image classification is acknowledged as a limited benchmark for task-driven capabilities. The paper positions this approach as both biologically inspired and more computationally efficient, reviving older attention-based vision ideas with modern architectures.

Position: Vision encoders should be image size agnostic and task driven.

**Strengths:**

1. The paper is well written, clear in communication and its stance

2. Biological inspiration put across is valid and the efficiency argument is well motivated.

3. The PoC further adds well in support of the argument.

4. The proposed proof of concept and he paper make it clear that biological inspiration may not imply copying of the biological system as we may be limited by hardware and engineering potential.

5. The idea of learning the right patches does seem well motivated although may be expensive given we want to build task driven encoders.

6. good discussion about open questions

**Weaknesses:**

1. If we adopt separate task-driven vision encoders for different tasks, how do we scale their training efficiently? The potential efficiency gains at inference may be offset by the cost of training and maintaining multiple large parameter sets across tasks.

2. The paper does not provide concrete measurements of efficiency (e.g., FLOPs, latency, memory usage) for the representative task and pipeline, making it difficult to evaluate whether the proposed approach actually delivers computational savings.

3. In domains with limited labeled or unlabeled data, how would we train effective task-driven encoders? The paper does not address whether the policy and encoder could generalize to low-data settings or transfer from other domains.

4. The paper does not discuss how the proposed approach would extend to other learning paradigms, such as vision–language models (VLMs) or broader multimodal systems. In such settings, integrating a task-driven, size-agnostic encoder may require adapting the pipeline to work with text or other modalities, and it is unclear whether the two-stage process (policy training followed by model training) would remain feasible or become prohibitively complex and expensive.

**Questions:**

1. How would we scale training of different vision encoders for different tasks?
2. In domains with limited labeled or unlabeled data, how would we train effective task-driven encoders?
3. How would the sample pipeline work in a VLM based setting?

**Alternative Position:**

Yes, and alternative positions are well-considered and named but not addressed

**Author Identification:**

Yes, multiple of the authors.

**Context:**

3

**Discussion:**

3

**Ethics:**

["NO or VERY MINOR ethics concerns only"]

**Position:**

Yes, the paper argues for or against a position related to machine learning.

**Support:**

3

**Thoroughness:**

4

---

### Note · Authors · 2025-08-22

**1-10 Additional Comments:**

We noticed that according to the guidelines in the call for position papers, the reviewers do not have to agree with the stated position. We've observed that the 'Weaknesses' section frequently includes counter-arguments to the paper's stated position. Personally, we think this is positive for position papers, because it shows the potential for discussions in the community.

**1-11 Submit Again:**

Definitely yes

**1-1 Submission Process:**

4

**1-2 Next Year:**

We like the idea of the position paper track. The main thing we would like for next year is to have the process more clearly defined beforehand - the timeline and whether there is a rebuttal or discussion.

**1-3 Future Development:**

Not sure how this could be implemented. However, having some form of public discussion on the papers while they are still under review might give a strong signal about the potential of meaningful community debates on the position of the paper. This might be a very good signal for whether the paper is worthy of publishing.

**1-4 Interest:**

["Workshops for developing position papers"]

**1-5 Thoughtful:**

7

**1-6 Supportive:**

7

**1-7 Technical Aspects Versus Position:**

4

**1-8 Gate Keeping:**

10

**1-9 Camera Ready Changes:**

* __Better clarify__ that by task-driven we do not mean to have many encoders specialized for different tasks, but **a single generalist encoder that can be prompted for many tasks** and perform dynamic computation based on the task.
* __Add a bit of discussion about the potential application for Vision Language Models (VLMs)__.  In our current version, we didn't include how our approach can be adapted to the more advanced VLMs. Ideally, we believe it is possible and practical. As questioned by Reviewer-X9ha, we are happy to include this discussion.

**3-1 Review Response1:**

X9ha

**3-2 Reaction To Review1:**

**W1&Q1: Scaling training for multiple tasks**: We have to clarify that the reviewer has a misunderstanding: we are **NOT** for building separate encoders for different tasks, but **a single encoder capable of dynamic task-dependent computation** —spending less compute on easy tasks and more on challenging ones. The system has two components: a transformer and a policy (for where to look next). The transformer will be pretrained (self-supervised) like current vision encoders (e.g., DINO) to be able to handle many tasks generally. The policy would arguably require task-specific fine-tuning. The pretraining of the transformer should address the reviewer’s concerns.

**W2: Efficiency measurements**: Our experiments are preliminary proof-of-concept. We will include comprehensive technical details when ready for the main track.

**W3&Q2: Generalization to low-data**: As mentioned in W1&Q1, the transformer is pretrained to be general like the current SOTA encoders (DINO). This should help with generalization on low-data tasks.

**W4&Q3: Extend to VLMs**: With VLMs an image encoder extracts visual features that are downstream used by an LLM. The features extracted by our proposal can similarly be aligned and used with text tokens. Architecturally, it is possible and practical to incorporate ours into VLMs. This would require extending the policy to be conditioned not only on the transformer’s state, but also on the text prompt. Additionally, the current vision encoders used by VLMs extract features in a static way. However, we believe it is good to dynamically extract the most relevant visual features according to the current text prompt, which could be tackled by our proposal of a task-driven model.

**3-3 Review Response2:**

mgPM

**3-4 Reaction To Review2:**

We thank the reviewer for recognizing the merits of our proposed position. We note that the mentioned challenges (occlusions, lighting variations, background clutter, object poses) are orthogonal to our core proposal. We should better clarify in the manuscript that our goal is to have a single generalist model that is able to solve many tasks (with dynamic computations as required by the given task). As mentioned in Section 7 (Open questions), we aim to pretrain the transformer in a self-supervised manner on a large scale of unlabeled images. This should be able to handle the majority of the challenges being mentioned.

**3-5 Review Response3:**

peVm

**3-6 Reaction To Review3:**

**W1: Disagreement on Task-Driven Position**: The reviewer’s counter argument is already discussed in L347-356 (Alternative views). While we agree that the current task-agnostic models are easier to use, we believe task-driven encoders will be eventually more efficient. Most importantly, the reviewer does not have to agree with the stated position.

**W2&Q: More Evidence**: We want to clarify that biological motivation serves simply as a point of inspiration - we do not try to copy biological vision in any structural way. In Section 3, we provide examples (1) “finding Waldo” game and (2) the game “ask questions after the image is hidden”. They illustrate the task-driven nature of human vision. We can further add reference to a work demonstrating the task-driven nature of human vision [1].
[1] The Anticipatory and Task-Driven Nature of Visual Perception. Sebo Uithol, et al.

**W3: Encoder being ViT**: We respectfully clarify a misunderstanding, possibly from L182. Unlike ViTs, we explicitly reject fixed-size patches that scale poorly with image size. While we employ BERT-like transformers (similar to ViTs) and multiple learned state tokens (similar to DINO with registers), our approach fundamentally differs. We propose to use 1) top-down multi-zoom patches; 2) the transformer iteratively while evolving internal state and not requiring backpropagation through time. Both of these make us very different from ViTs.

**W4: On Experimental Results**: Our experiments on ImageNet-1K only work as proof-of-concept, to show the feasibility of our ideas: 1) the transformer can handle multi-zoom patches, and 2) it is possible to train an iterative transformer evolving internal state without backpropagation through time. A proper baseline requires self-supervised pretraining and is a work in progress.

---

### Meta-Review · Area_Chair_iLHC · 2025-09-10

**Rating:** 8
**Confidence:** 4

**Strengths:**

Reviewers found the paper to be well-written, well-motivated, interesting and relevant

**Weaknesses:**

The paper led some readers to be confused about the specific details of the position, for example around the idea of task-specific encoders. There was also a lack of discussion of concrete measures of efficiency.

**Questions:**

How would policies be fine-tuned in data-limited settings? How could this work with VLMs?

**Thoroughness:**

3

---

### Decision · Program_Chairs · 2025-09-26

Reject